# Sinterability and Dielectric Properties of LiTaO_3_-Based Ceramics with Addition of CoO

**DOI:** 10.3390/ma13071506

**Published:** 2020-03-25

**Authors:** Youfeng Zhang, Yali Yao, Shasha He

**Affiliations:** School of Materials Engineering, Shanghai University of Engineering Science, Shanghai 201620, China; yao2864513359@163.com (Y.Y.); He13100619279@163.com (S.H.)

**Keywords:** LiTaO_3_ ceramics, cobaltous oxide, microstructure, dielectric properties

## Abstract

Lithium tantalite (LiTaO_3_) is a common piezoelectric and ferroelectric crystal, but the LiTaO_3_ polycrystalline ceramics have rarely been reported, and their refractory character presents difficulties in their fabrication. In this study, LiTaO_3_-based ceramics with different amounts of CoO were prepared by pressureless sintering at 1250 °C, and the effects of the amount of sintering aid on the sinterability, microstructure, and dielectric properties of the ceramics were investigated. The relative densities of the LiTaO_3_-based ceramics were significantly improved by the addition of CoO powder. The LiTaO_3_-based ceramics achieved the highest relative density (89.4%) and obtained a well-grained microstructure when the added amount of CoO was 5 wt.%. Only the LiTaO_3_ phase in the ceramics was observed, indicating that the ions Co diffused into the LiTaO_3_ lattices and mainly existed in two forms: Co^2+^ and Co^3+^. The effects of the added amount of CoO on the dielectric properties of the LiTaO_3_-based ceramics were studied thoroughly. Consequently, the dielectric constant was enhanced, and the dielectric loss decreased in the LiTaO_3_-based ceramics with the addition of CoO. The optimal value was obtained at 5 wt.% of CoO-added LiTaO_3_-based ceramics.

## 1. Introduction

Lithium tantalite (LiTaO_3_), an excellent single crystal, has high planar electromechanical coupling, a mechanical quality factor, and low acoustic transmission loss and is used in various applications because of its low dielectric constant [1,2,3,4]. Many previous studies have focused on single crystal LiTaO_3_ [5,6,7,8]. However, relatively little attention has been paid to the fabrication and sintering of LiTaO_3_ ceramics or their microstructures and properties, because LiTaO_3_ is beset with difficulties regarding densification in the sintering process [9,10,11,12,13]. Previous studies have tried to facilitate the sintering of the LiTaO_3_ ceramics by adding oxides/fluorides or by doping with two steps of powder synthesis first, followed by sintering [14,15]. Meanwhile, there have been many studies on LiTaO_3_ solid solution systems such as Mg, CaTiO_3_, Ag, Cu-doped and LiF, MgF_2_ co-doped LiTaO_3_ systems [16,17,18,19,20]. Nowadays, lead zirconate titanate (PZT) is the most widely-used piezoelectric ceramic, but the production and use of lead-containing materials will produce serious lead volatilization, which is not conducive to environmental protection, and the toxicity of lead oxide is very large. Therefore, in the field of electronic ceramics, even though the piezoelectric properties of lead-free piezoelectric ceramics are less than those of lead-containing piezoelectric ceramics, many lead-free ceramic materials have been developed to replace these lead-based ceramic materials in recent years [21,22]. The research on LiTaO_3_-based piezoelectric ceramics is valuable for the application in potential engineering and is expected to lead to the development of a new lead-free piezoelectric ceramic. However, the sintering process, microstructures, and properties of LiTaO_3_-based ceramics are yet to be investigated systematically.

There are three main reasons why it is difficult to fabricate high-density LiTaO_3_-based ceramics. First, sintering densification is accompanied by grain growth, and an increased sintering temperature leads to abnormal grain growth in the ceramic, resulting in residual pores at grain boundaries [15]. Second, due to the crystal structure of LiTaO_3_, it has significant crystallographic anisotropy in its coefficients of thermal expansion; this causes tremendous stress during cooling, thereby making it challenging to sinter a dense LiTaO_3_ ceramic [23]. Third, slight volatilization of Li_2_O at higher sintering temperatures (over 1300 °C) causes Li deficiency [24]. Therefore, preparing a dense LiTaO_3_-based ceramic requires an efficient sintering aid. Co is a multi-valence element that can improve the performance of many functional ceramics. Accordingly, researchers have shown that Co doping can improve the dielectric properties of ceramics [25,26,27,28]. In this paper, CoO was added to fabricate the LiTaO_3_-based ceramics through a pressureless sintering process using raw CoO and LiTaO_3_ powders. The LiTaO_3_-based ceramics were successfully fabricated herein by adding different amounts of CoO. The effects of the different components on the sinterability and dielectric properties of the LiTaO_3_-based ceramics were studied.

## 2. Experimental Details

LiTaO_3_-based ceramics with different amounts of CoO (mass fractions of 0, 1, 3, 5, and 7 wt.%, referred to as 0CLT, 1CLT, 3CLT, 5CLT, and 7CLT, respectively) were fabricated by a pressureless sintering method. Commercially available LiTaO_3_ powder (Fangxiang Industry Co. Ltd., Shanghai, China) and CoO powder (Sinopharm Chemical Reagent Co. Ltd., Shanghai, China) were used as the raw materials. On the basis of the mass ratio, the two powders were weighed and then ball-milled in alcohol with carnelian balls for 24 h, after which the slurry was stirred and dried to obtain CoO/LiTaO_3_ (CLT) composite powder. The composite powder was mixed with a 7 wt.% polyvinyl alcohol binder, pressed into discs with diameters of 16 mm and thicknesses of approximately 2 mm and calcinated at 800 °C for 2 h. Finally, each disc was sintered at a sintering temperature of 1250 °C for 3 h. The relative density of each sintered ceramic was calculated on the basis of the sample volume and mass; the theoretical density was approximately 7.45, 7.44, 7.42, 7.39, and 7.37 g/cm^3^ for 0, 1, 3, 5, and 7CLT, respectively. The crystalline structure was obtained using an X-ray diffractometer (X’Pert PRO; PANalytical, The Netherlands) using Cu Kα radiation. The surface element compositions and the chemical states of the samples were analyzed through X-ray photoelectron spectroscopy (XPS, ESCALAB 250Xi, Thermo Fisher, USA) with Al Ka radiation. The microstructure of the ceramic was characterized using a scanning electron microscope (SEM) (S-3400N; Hitachi, Japan). The samples were plated by silver painting on both sides of the polishing pellets and then kept warm at 850 °C for 10 min to characterize the dielectric properties. The frequency-dependent dielectric constant and the dielectric loss were obtained at room temperature from 100 Hz to 1 MHz using an impedance analyzer (Model Agilent E4990A, Central South University, Hunan, China). The temperature dependence of the dielectric constant and the dielectric loss was measured by an LCR meter (Model HP4284A, Agilent Technologies Ltd., Hyogo, Japan) at 10 kHz from room temperature to 750 °C.

## 3. Results and Discussion

The relative densities of the LiTaO_3_-based ceramics of 1, 3, 5, and 7CLT sintered at 1250 °C are given in Figure 1. The relative density of all samples obviously increased at first and then decreased subsequently with an increasing amount of CoO; the maximum value (89.4%) was obtained at 5CLT. As a sintering aid, CoO can efficiently accelerate the sintering process of the LiTaO_3_-based ceramics and promote densification, but the addition of excessive CoO decreased the density of ceramics, which could be related to the inhomogeneity of the grain growth during the sintering process. Yao and Li et al. also reported similar results when studying the effect of MnO_2_ on the LiTaO_3_ and (Bi_0.5_Na_0.5_)_0.94_Ba_0.06_TiO_3_ ceramics [29,30].

Figure 2 shows the XRD patterns of the LiTaO_3_-based ceramics with different amounts of CoO. It can be clearly seen that only one phase of LiTaO_3_ exists in the LiTaO_3_-based ceramics with different amounts of CoO added, and this was indexed well with a trigonal (rhombohedral) crystal structure (JCPDS card number: 29-0836). Furthermore, a secondary phase was not found in samples, which means that CoO probably diffused into the LiTaO_3_ lattice to form solid solutions, and the crystal structure of LiTaO_3_ was not significantly changed in the LiTaO_3_-based ceramics with the addition of CoO. The details of the XRD results can be observed in Figure 2b, which shows a partial main diffraction peak of the patterns of the LiTaO_3_-based ceramics. Figure 2b shows that the main diffraction peak moved towards the left (low angle direction) with an increase in the added amount of CoO. This result supports the prediction that the Co ions would diffuse into the LiTaO_3_ lattices, thereby leading to an expansion of the lattice. Since the radius of Co^2+^ is 0.065 nm (in low spin states) and 0.0735 nm (in high spin states), the radius of Co^3+^ is 0.0545 nm (in low spin states) and 0.061 nm (in high spin states), and that of Ta^5+^ is 0.064 nm [31]. The replacement of the smaller Ta^5+^ by Co^2+^ cations caused lattice expansion of LiTaO_3_ and the diffraction peak shifted.

XPS studies were conducted in order to supplement the composition, electronic configuration, and surface state of the CLT ceramics. Co2p XPS spectra of LiTaO_3_-based ceramics with different amounts of CoO are shown in Figure 3. It can be seen from Figure 3 that the Co 2p spectrum consists of two wide peaks of Co 2p3/2 and Co 2p1/2, which are mainly ascribed to Co–O bonds. The different valence states of the doped Co ions were investigated by fitting overlapping Co 2p3/2 peaks. Mixed Co valence states (Co^2+^ and Co^3+^) in all samples were observed by XPS due to the oxidation of Co^2+^ ions. The peak of Co^2+^ ions appeared at a lower binding energy, while the peak of Co^3+^ ions appeared at a higher binding energy, revealing the co-existence of Co^2+^ and Co^3+^ ions. So, the Co ions mainly existed in two forms: Co^2+^ and Co^3+^. It can be seen that Co ions were mainly present in the form of Co^2+^ according to the fitting results of the Co 2p3/2 peak. The phase containing the Co element cannot be detected in XRD because of the small content in CLT ceramics.

The SEM micrographs of the LiTaO_3_-based ceramics with different amounts of CoO are shown in Figure 4. In Figure 4, it can be seen that the porosity decreased at first and then increased as the added amount of CoO increased. The sinterability of the sample with 1 wt.% of added CoO was the worst; it was porous and not densified, which agrees with the investigation of relative density. The pores were achieved minimally when the added amount of CoO was 5 wt.%. The grain size of LiTaO_3_ was uniformly distributed in the ceramic 5CLT because adding Co cations into the LiTaO_3_-based ceramic would create a replacement of Ta^5+^ by Co^2+^/Co^3+^, leading to oxygen vacancies. Since the radius of O^2−^ (0.140 nm) is much larger than the radii of other ions in LiTaO_3_-based ceramics, the sintering of LiTaO_3_-based ceramics is mainly restricted by the diffusion of oxygen ions [15]. Therefore, the occurrence of oxygen vacancies results in an easier grain boundary mobility, which is conducive to better sintering behavior. Combined with the XRD results, it can be seen that the second-phase particle morphology does not appear in LiTaO_3_-based ceramics with different amounts of CoO, which indicates that the Co ions may have dissolved completely into the LiTaO_3_ lattice. However, the addition of excessive CoO led to the worse sinterability for the LiTaO_3_-based ceramics.

Figure 5 shows the frequency-dependent dielectric constant and the dielectric loss of CLT ceramics as a function of the frequency in the range from 100 Hz to 1 MHz at room temperature. A higher dielectric constant value was presented at a low frequency, and then the permittivity tended to have a stable value at frequencies over 10 kHz, as shown in Figure 5a. At the same frequency, the permittivity of CLT ceramics increased initially, and then decreased with an increase in the amount of CoO. Subsequently, a maximum value of about 50 was achieved when the CoO amount was 5 wt.%; this is similar to that of a single crystal LiTaO_3_. The dielectric loss of CLT ceramics with different amounts of CoO at room temperature are presented in Figure 5b. It can be seen that the dielectric loss of all samples decreased with an increase in the test frequency. The variation in the dielectric loss curves was contrary to the dielectric constant, i.e., it decreased firstly and then increased with an increase in the amount of CoO at the same frequency. The minimum value was obtained when the added amount of CoO was 5 wt.%. The largest dielectric constant and the smallest dielectric loss were gained when the addition of CoO was 5 wt.% in the LiTaO_3_-based ceramic. It is well known that many factors can affect the dielectric properties of ceramics, for example, porosity, second phase, and ionic polarizability [32]. The porosity gradually decreased when CoO was added, according to the SEM photographs and the curves of the relative density. A fine microstructure was gained when the addition of CoO was 3 or 5 wt.%, which resulted in improvement of the dielectric constant. The dielectric constant decreased and the dielectric loss increased when more than 5 wt.% CoO was added, due to the worsening sinterability.

The temperature-dependent dielectric constant (ε_r_) and dielectric loss (tanδ) of the LiTaO_3_-based ceramics with different amounts of CoO were measured at 10 kHz as a function of temperature from room temperature to 750 °C, and the results are shown in Figure 6. The maximum dielectric constant at 0CLT was 106; the maximum value of the dielectric constant increased by up to 3055 at 5CLT and then decreased as the added amount of CoO increased. It can be seen from Figure 6a that there was a peak value at the phase transition point (Curie point T_m_) of ferroelectric to paraelectric in LiTaO_3_-based piezoelectric ceramics with an increase in temperature, followed by a slow decrease. The dielectric constant will be unusual when the dielectric changes in phase transition or other microstructures. The dielectric constant was lower when the temperature was lower than 300 °C because the relaxation of polarization takes a long time. With an increase in temperature, the polarization is established more fully, which results in the dielectric constant increasing gradually. Simultaneously, an increase in temperature prevents thermal movement that is against the mass. The regular motion of the mass point hinders an increase in polarization, and the corresponding dielectric constant decreases. This means that the motion of atoms in the system is intensified with an increase in temperature, which hinders the motion of particles participating in polarization and leads to a decrease in polarizability. Therefore, the dielectric constant peak value will appear at a certain temperature, i.e., at the Curie point. With an increase in the amount of CoO, the dielectric peak becomes wider, indicating that a large amount of CoO can induce phase transformation dispersion near the Curie point. This is mainly because the addition of CoO increases the disorder of cations and the volatility of the composition. In the LiTaO_3_ ceramic system, Li is a volatile element, so the more Co ions that are dissolved into the LiTaO_3_ lattice, the greater the disorder degree of the whole system and the faster the volatilization of Li ions. Figure 6b shows the temperature-dependent dielectric loss of LiTaO_3_-based piezoelectric ceramics with different amounts of CoO at 10 kHz. The dielectric loss of all samples increases with an increase in the temperature and a peak appears near the Curie temperature. A relatively small value of dielectric loss was obtained for the LiTaO_3_-based ceramics with the addition of 5 wt.%, as shown in Figure 6, which is related to the relative density of the LiTaO_3_-based ceramics.

For relaxor ferroelectrics, Uchino et al. proposed a modified Curie–Weiss law to describe the degree of phase transformation dispersion of relaxor ferroelectrics [33,34,35].
(1)$$\frac{1}{\varepsilon_{r}}-\frac{1}{\varepsilon_{m}}=\frac{{\left(T-T_{m}\right)}^{\gamma }}{C}$$
where ε_m_ is the maximum dielectric constant, T is the test temperature, T_m_ is the temperature of the ε_m_, C is the Curie–Weiss constant, and γ is associated with the diffuseness degree. γ = 1 indicates that the normal phase transition satisfies the Curie–Weiss law; γ = 2 indicates the complete dispersion phase transition. The logarithm is taken on both sides of Formula (1-1) to get Formula (1-2):(2)$$\log(\frac{1}{\varepsilon_{r}}-\frac{1}{\varepsilon_{m}})=\gamma \log\left(T-T_{m}\right)-\mathrm{logC}.$$
y = Ax + b was used for the simulation, and the value of γ was obtained by linear fitting of data points. Figure 7 shows the variation in $\log(1/\varepsilon -1/\varepsilon_{m})$ with $\log(T-T_{m})$ at different components and 10 kHz. It can be seen from Figure 7 that there was a linear relation, and the value of changed from 1.0 to 1.30, and it was close to 1.0 when the amount of CoO added was 3 wt.% or 5 wt.%, i.e., the ceramics presented normal ferroelectric behavior. Due to the different valence and ionic radii, ion substitutions of Co^2+^ or Co^3+^ for Ta^5+^ at B sites generated chemical and displacive disorder in the LiTaO_3_ lattice [36]. This would lead to a lattice with a deviated ferroelectric phase transition, but the diffuse phase transition phenomenon in LiTaO_3_ ceramics was not obvious. A similar influence of a new disordered structure was reported for an Mn-doped Pb-based ferroelectric ceramics system [37].

## 4. Conclusion

LiTaO_3_-based ceramics with different amounts of CoO were successfully fabricated by pressureless sintering at 1250 °C. The effects of different amounts of CoO on the sinterability, microstructures, and dielectric properties of the CLT ceramics were investigated. The sinterability of the LiTaO_3_-based ceramics was improved by increasing the amount of CoO added, and the relative densities of the LiTaO_3_-based ceramics were significantly enhanced. The LiTaO_3_-based ceramics achieved the highest relative density (89.4%) and obtained well-grained microstructures when the amount of CoO added was 5 wt.%. Only the LiTaO_3_ phase was observed in the ceramics, indicating that the Co ions diffused into the LiTaO_3_ lattices and mainly existed in two forms: Co^2+^ and Co^3+^. At room temperature, the dielectric constant of the LiTaO_3_-based ceramics increased first in the frequency range from 100 Hz to 1 MHz, and then decreased with an increase in the amount of CoO at the same frequency. It also achieved the maximum value of 50 when the CoO content was 5 wt.%. Similarly, the dielectric loss was the lowest when the added CoO was 5 wt.%. The maximum dielectric constant gradually increased up to 3055 at 5CLT and then decreased. The variation tendency of the dielectric loss versus the component was consistent with the dielectric constant in the temperature range of room temperature to 750 °C. Therefore, the highest relative density, a well-grained microstructure, and better dielectric properties were obtained with LiTaO_3_-based ceramics with 5 wt.% added CoO.

## Figures and Tables

**Figure 1 materials-13-01506-f001:**
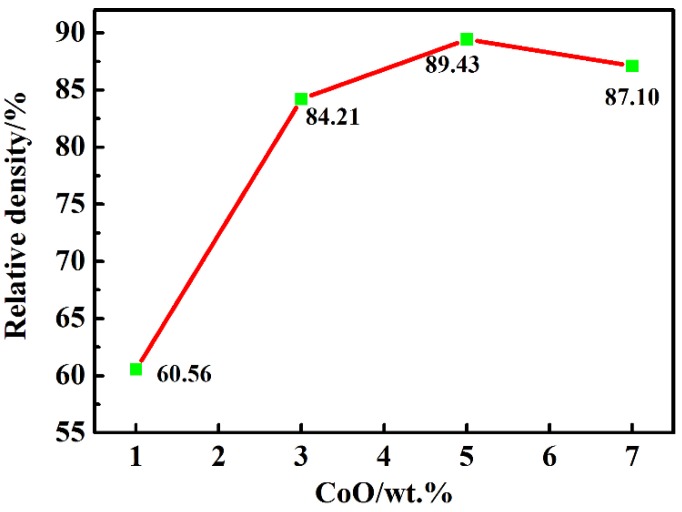
Relative density of LiTaO_3_-based ceramics with different amounts of CoO.

**Figure 2 materials-13-01506-f002:**
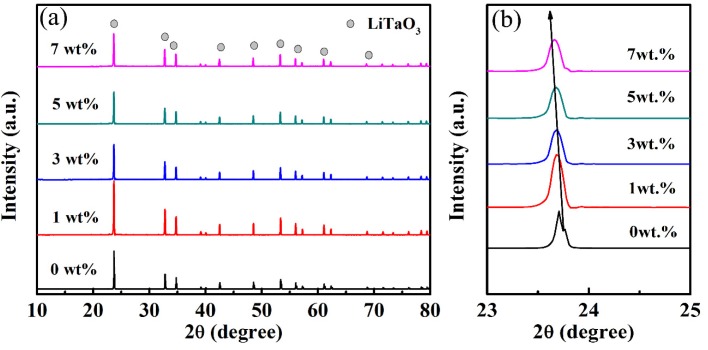
XRD patterns (**a**) and partial enlarged patterns of CoO/LiTaO_3_-based ceramics (**b**).

**Figure 3 materials-13-01506-f003:**
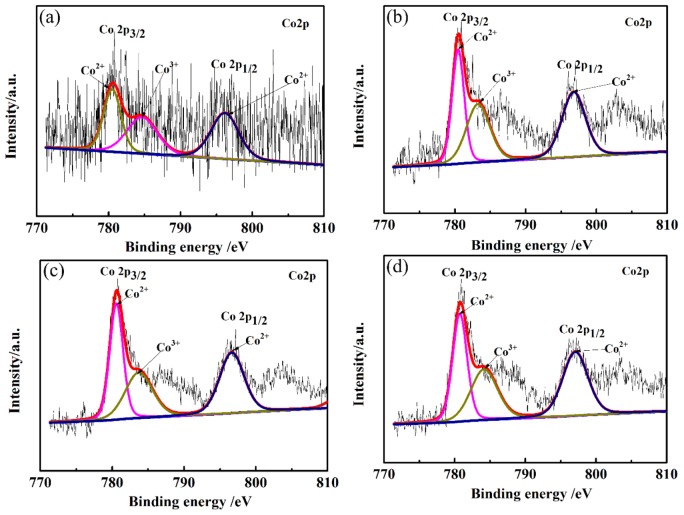
Co2p XPS spectra of LiTaO_3_-based ceramics with different amounts of CoO. (**a**) 1 wt.% (**b**) 3 wt.% (**c**) 5 wt.% (**d**) 7 wt.%.

**Figure 4 materials-13-01506-f004:**
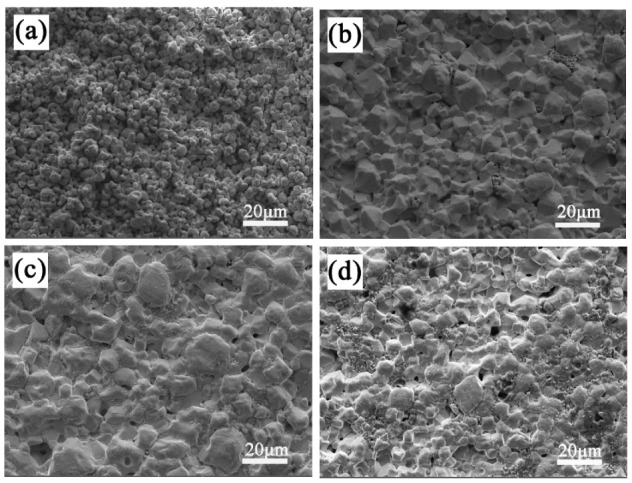
SEM micrographs of CoO/LiTaO_3_-based ceramics. (**a**) 1 wt.% (**b**) 3 wt.% (**c**) 5 wt.% (**d**) 7 wt.%.

**Figure 5 materials-13-01506-f005:**
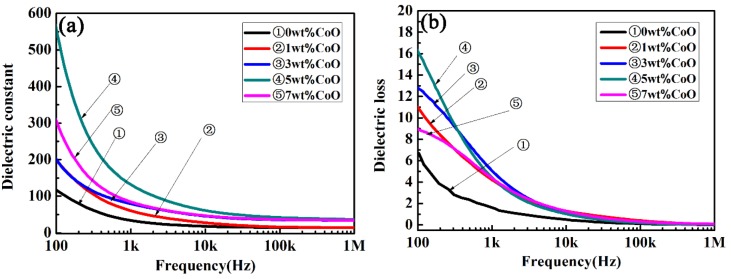
The frequency-dependent dielectric constant (**a**) and dielectric loss (**b**) of the LiTaO_3_-based ceramics at room temperature.

**Figure 6 materials-13-01506-f006:**
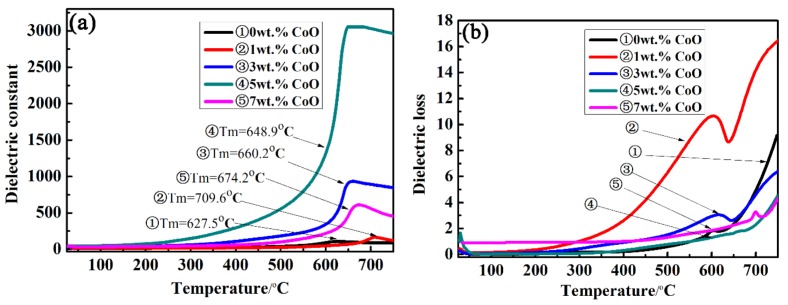
Dielectric constant (**a**) and dielectric loss (**b**) of LiTaO_3_-based ceramics with different amounts of CoO doping as a function of temperature at 10 kHz.

**Figure 7 materials-13-01506-f007:**
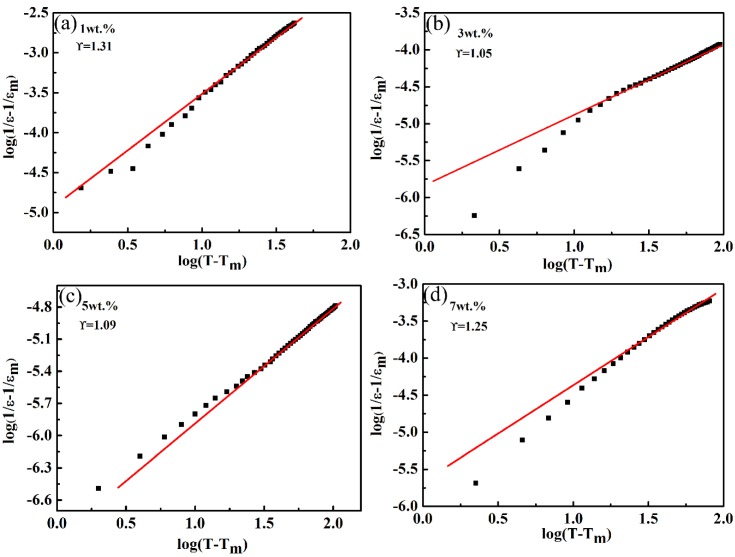
The variation of log (1/ε-1/ε_m_) versus log (T-T_m_) for the LiTaO_3_-based ceramics with different compositions at 10 kHz. (**a**) 1 wt.% (**b**) 3 wt.% (**c**) 5 wt.% (**d**) 7 wt.%.

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
