# Peer review of "Sinterability and Dielectric Properties of LiTaO3-Based Ceramics with Addition of CoO"

_materials, 2020, doi:10.3390/ma13071506_

Round 1

Reviewer 1 Report

Dear authors, please consider the following corrections, additions and explanations to your manuscript:

  1. Some parts of the manuscript are written in good English, but others do need major English language revisions, especially the chapter Results and discussion, for example: Line 83, 87, 89, 90, 92, 93, 96, 105, 112, 119, 122, 123,136, etc. etc.).
  2. There isn't any mention in the manuscript of the lithium tantalite ceramics' advantage, that it is lead-free (environmental impact).
  3. Could you add some comments and references on how this relatively low relative density (89.43%) could be improved by using advanced sintering processes.
  4. You mention “pressure-less sintering” in the abstract only (Line 13), while in the manuscript body there is “conventional sintering” (Line 54). Make it clear to the readers that these terms mean the same, please.
  5. There is no need (and it is confusing) to put the abbreviations for samples (1CLT, 3CLT, 5CLT and 7CLT) in the summary, as these designations were not used in any of the figures or figure captions. I suggest moving this referral to the chapter “Experimental details”. Also, pay attention that the textual explanation of the abbreviation “CLT” appears when the abbreviation first appears (see lines 54/58).
  6. You used “carnelian balls“ for ball-milling. Could you explain why this type of balls was selected, please? Is there a possibility of contamination of the homogenized LiTaO3 and CoO powders by the silicon dioxide (with possible inclusions) from the balls during the ball-milling?
  7. Line 62: “relative density of each sintered ceramic was calculated on the basis of the sample volume and mass”? Relative density is calculated on the basis of measured and theoretical density. Firstly, why did you calculate the sample density by using the fair method of “mass-over-volume”, instead of the Archimedes method? Secondly, there isn’t any information given on the theoretical density of used ("composite?") ceramics, which is needed to calculate the relative density of the sample.
  8. Which instrument was used for the XPS analysis?
  9. Use full name or full chemical composition instead of “BNBT” (Line 80), especially since this is not a ceramics-only journal, please.
  10. Is it a “composite ceramics” after all? There wasn’t any second phase visible (i.e. no dispersed second constituent), nor was it detected by XRD. The Co-ions have replaced the Ta-ions in the lattice itself.
  11. Add the XRD of pure LiTaO3 (0% CoO) if possible.
  12. Add in figure caption of Fig. 2 the description what is (a) and what is (b).
  13. Add in figure caption of Fig. 3 the description what is (a) to (d); use lower case for "Energy" on the x-asis title.
  14. Pay attention to unify all figure captions as in the MDPI paper template, considering the way of separating figure references (a) to (b) or (d), using colon, semi-colons, bold font and paragraph alignment.
  15. I suggest replacing the scale bars and the letters “20 um” in Fig. 4 from black to white to make the images more legible.
  16. Consider adding different data point symbols to curves in Fig. 5 and Fig. 6 to improve the readability of images in B&W printout. Or identify individual curves with tags (a-b-c-d).
  17. Add in Fig. 6 caption that the curves were obtained at room temperature.
  18. Line 149: “increasingly increased”?; CLT, not MLT; “then decreased with increasing of the amount of CoO” --> with increasing the temperature.
  19. Please, explain this better: “temperature increasing busts thermal movement that is against with the mass”.
  20. How was the Curie temperature measured? Also, add values of Curie temperatures to the chart in Fig. 6.
  21. Correct the entire caption of Fig. 6, as there aren’t images a-d in this figure and a-b are two property charts; Fig6. 6 b, y-axis: “Loss” not capitalized; “10KHZ” to "10 kHz".
  22. Correct the left side of equation (1-2), as it should be after applying the logarithm to equation (1-1).
  23. Describe epsilon, T, C and Tm from equation (1-1) as well.
  24. Either all or none of the variables should be in italics (check line 175, 176, 146). The MDPI paper template doesn't use italics for the variables.
  25. Decadic logarithm (log) was used in Fig. 7, in the Fig. 7 caption and in the text (Line 177), so the natural logarithm (ln) in expression (1-2) should also be log.
  26. Fig. 7 caption: add the description what is (a) to (d); correct the fonts of variables; add “at 10 kHz”; (variables in these carts were not formatted in italics).
  27. Finally, there are numerous typos like: missing spaces, semi-colons instead of commas, incorrect symbols for centigrade, wt% and wt.%, KHz, etc.

Reviewer 2 Report

The present work “Sinterability and dielectric properties of LiTaO3-based ceramics with addition of CoO” by Y. Zhang et al. is an interesting study of the use of CoO in order to produce dense ceramics and increase the dielectric properties of LiTaO3 ceramics. It could be useful to researchers in piezoelectric materials. My opinion for the present work is that some conclusions and explanations should be clarified and discussed before its publication on the “Materials” journal.

For instance:

  1. Line 91 and 117. Authors should provide the reference from which they obtained the ionic radius. It is an important point of view because they are related the shift in XRD due to the substitution of Ta by Co, and especially, if they are indicating that the two oxidation states of cobalt are presented.
  2. Line 105: The sentence “Co3+ cannot be detected in XRD because the small content in CLT ceramics” should be clarify. Of course, XRD does not detect oxidation states, so this conclusion should be explain better.
  3. Line 162: The final sentence “…and the volatility of composition” should be explain in more detail. What does it means?
  4. In my opinion, the last figure (Fig 7) and the explanation of these figures does not provide any conclusion and additional interesting information to the work. I would proposed to eliminate this figure and the text form line 169.
  5. However, I would include a discussion or comparison of the results obtained in this paper with others observed in the literature. For example, the dielectric loss or the dielectric constant values could be compared.
  6. Could authors included any piezoelectric parameter? It could be very interesting for this system and in particular for the researchers in the area of piezoelectric materials.

Minor changes

  1. XRD: JCPDS-ICDD number of the pattern used for the identification of LiTaO3 phase should be provide.
  2. XRD: The quality of Fig 2(a) should be improved. For example, peaks seems to be lines.
  3. Line 151: Tc value for the LiTaO3 without cobalt should be provided.

 I hope these comments be helpful and serve to improve this study.

Thank you very much.

Yours sincerely

Round 2

Reviewer 1 Report

Dear authors,

Thank you for your kind responses and for all the modifications and additions to the manuscript. I am sure your paper will now be much more informative and easier to read.

******

Reply:Thank you for your circumspection. We have modified the term “composite ceramics” to “-based ceramics” in the whole article.

A minor remark by the reviewer: The old term “composite” still appears in Line 19 (Abstract) and in Line 220 (Conclusion).

******

Ambiguity of Tc and Tm:

As you had previously well explained in the manuscript, as well as in your reply, it looks as the temperature Tc is equal to temperature Tm.

You have explained in detail on page 6 what Tc is, and then, on page 7, you have introduced a new variable, Tm (in Fig.6, eq. (1-1) and below). But there isn’t any comment on the connection between these two variables.

Similarly, Lines 174 and 175 suggest to the reader to look for Tc values in Fig.6. But Fig.6 does not present Tc, but Tm (confusing).

I do suggest that you somehow connect the narrative of these two variables and highlight the differences (or similarities) between these two terms more clearly.

Very best regards,

Reviewer 2 Report

Dear authors

Thank you very much for your changes and the detail response to my suggestions. I have revised your manuscript again and I consider that it could be published in the Materials journal in the present form. 

I would simply like to add that, although it is not necessary for this manuscript, it would be convenient to include piezoelectric studies of this material in a future research.

I hope my comments have served to improve the quality of the manuscript.

Yours sincerely
